# Co-Occurrence of EBV-Positive Mucocutaneous Ulcer (EBV-MCU) and CLL/SLL in the Head and Neck Region

**Patricia Bott** [1,*], **Ilske Oschlies** [2], **Andreas Radeloff** [1] and **Maureen Loewenthal** [1]

1   Division of Otorhinolaryngology, Head and Neck Surgery, University of Oldenburg, Steinweg 13-17, 26122 Oldenburg, Germany; andreas.radeloff@uni-oldenburg.de (A.R.); maureen.loewenthal@evangelischeskrankenhaus.de (M.L.)
2   Hematopathology Section and Lymphnode Registry, Department of Pathology, University Hospitals Schleswig-Holstein, Christian-Albrecht-University, Arnold-Heller-Str. 3/14, 24105 Kiel, Germany; ioschlies@path.uni-kiel.de
*   Correspondence: patricia.bott@web.de

**Abstract:** EBV-positive mucocutaneous ulcer (EBV-MCU) was classified as a rare new entity of the lymphoproliferative B-cell diseases by the WHO in 2017 and must be distinguished from head and neck squamous cell carcinoma by early biopsy. The aim of the study is to raise awareness of the disease and to give a review of the current literature and a recommendation for EBV-MCU management. All EBV-MCU cases of the head and neck region published so far were included. We also report a case of a pharyngeal EBV-MCU in an 89-year-old patient who was immunosuppressed by chronic lymphatic leukaemia/small lymphocytic lymphoma (CLL/SLL). In contrast to all previously described cases, histopathology showed a co-infiltration of EBV-MCU and CLL/SLL. A total of 181 cases were identified on PubMed and summarised. EBV-MCU was predominantly caused by immunosuppressive drug therapy. Complete remission could be achieved in 68% of cases and was mainly attributed to a reduction of the immunosuppressive therapy alone (72%). However, some severe cases require more aggressive treatment. Regarding the various histopathologic similarities to other lymphoproliferative disorders, the diagnosis of EBV-MCU can be misleading, with a great impact on patient care and treatment. This diagnosis must be made with caution and requires a combination of clinical, morphological and immunophenotypic features.

**Keywords:** Epstein–Barr virus; mucocutaneous ulcer; head and neck ulcer; EBV-positive mucocutaneous ulcer

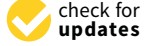



## 1. Introduction

The Epstein–Barr virus (EBV, human gammaherpesvirus 4)-positive mucocutaneous ulcer (EBV-MCU) was first described in 2010 by Dojcinov et al. and was recognised as a unique entity of the lymphoproliferative B-cell disorders by the World Health Organization (WHO) in 2017 [1,2]. Since then, about 200 cases have been described in the literature worldwide. However, so far, EBV-MCU still remains an under-recognised entity in the head and neck region to most otorhinolaryngologists and maxillofacial surgeons. Therefore, the aim of this study is to raise the awareness of EBV-MCU as a differential diagnosis for ulcerations in the head and neck region, provide them with an overview of therapy options and avoid further mis- or overtreatment of patients.

EBV-MCU clinically presents itself as a shallow and sharply circumscribed ulcer that is reported to be mostly located in the oropharyngeal mucosa (52%), on the skin (29%) or the gastrointestinal tract (19%) [3]. These ulcers generally occur as a solitary lesion but have been reported to be multifocal in 17% of all cases. Extended and painful oral and oropharyngeal mucosal defects can lead to eating disorders resulting from odynophagia and dysphagia and to severe weight loss. Roberts et al. reported a case of a 49-year-old woman with an extended oral EBV-MCU eroding the maxilla and palate, with the

spontaneous loss of multiple maxillary teeth, increasing pain, the inability to tolerate food and a 36 kg unintentional weight loss, finally leading to aspiration, pneumonia and sepsis that had to be treated with radiation therapy after a failed therapy with rituximab and had a duration of response of at least 6 months [3]. Au et al. reported a case of a 72-year-old woman with a long history of Crohn's disease presenting with a 6-month history of a base of the tongue ulcer and severe odynophagia requiring gastrostomy tube placement [4]. Alternatively, as recently published by Li et al., complete healing of a deep ulcerating EBV-MCU of the oral cavity was only achieved after bone sequestrectomy followed by closure with a local mucosal flap [5].

EBV-MCU localised in the gastrointestinal tract can also cause a variety of abdominal symptoms, from appetite and weight loss to obstruction or even abdominal emergencies, such as colon perforation [6]. Beside the painful local tissue erosion, which can be severe, systemic manifestations such as fever, lymphadenopathy, organomegaly or bone marrow involvement are normally absent [1,7–9].

The underlying pathomechanism of EBV-MCU evolvement is attributed to a primary infection with the EBV and a subsequent viral genome latency in B cells and certain epithelial cells. Most individuals are exposed to the virus during childhood and adolescence and undergo an asymptomatic infection or a self-limiting infective mononucleosis [10]. The oropharynx is the main access of entry for the virus and the localisation of primary infection, often within Waldeyer's ring, which explains why EBV-MCU mainly occurs on the oropharyngeal mucosa.

After a natural infection, the circular EBV deoxyribonucleic acid (DNA) persists as an episome in the cell nucleus of the infected lymphocytes through which the EBV latency within the B cells progresses in a unique and complex latency program. EBV-produced proteins transform naïve B cells into proliferating blasts that form germinal centres and later, after the restriction of further virus gene expression, induce the B cells to differentiate into long-lived resting memory B cells. Due to the lack of major histocompatibility complex (MHC) presentation of viral antigens, the infected cells miss T-cell detection and can avoid immunosurveillance. Through the activation and differentiation of the memory B cells into plasma cells and starting the lytic life cycle, it is possible for the virus to spread and infect other B cells [11].

During immunosuppression, which, in combination with a latent EBV infection, is the main pathophysiological mechanism for EBV-MCU development, the immunosurveillance by cytotoxic T cells is reduced, and the virus can enter lytic replication again. While the virus can be kept in a dormant state systemically, localised EBV-driven lymphoproliferations can appear.

Viral ribonucleic acid (RNA) and proteins can at least transform latently EBV-infected B cells into malignant cells by constitutional gene activation and the suppression of apoptosis [10,11].

EBV-associated malignant B-cell lymphoproliferations include diffuse large B-cell lymphoma (DLBCL), Burkitt lymphoma (BL), plasmablastic lymphoma (PBL) or lymphomatoid granulomatosis (LG). Some conventional histopathologic criteria overlap with those of an EBV-MCU and may lead to diagnostic challenges or even misdiagnosis [2,10–17].

There are various forms of immunosuppression that are attributed to the development of EBV-MCU. In most cases, EBV-MCU is iatrogenic (56%) and is caused by immunosuppressive drug therapy. Several cases related to azathioprine, cyclosporin A, cyclophosphamide, tacrolimus, mycophenolate, tumour necrosis factor-alpha (TNFα) inhibitors and, above all, methotrexate (MTX) as a treatment for rheumatoid arthritis have been described [3,14–20]. Interestingly, methotrexate and cyclosporin A seem to be able to directly activate EBV lymphocyte proliferation [3,19]. EBV-MCU has also been described in immunosuppressed recipients after solid organ or bone marrow transplantation [7,21,22].

The other great risk factor is immunosenescence (40%) due to advanced age caused by a natural depressed T-cell function [1]. Therefore, age-associated immunosenescence seems

to be a significant predisposing factor, especially for patients on immunosuppressive drug therapy [1,7,20,23–26].

EBV-MCU has been also attributed to primary immunodeficiencies and to human immunodeficiency virus (HIV) infections, but there are only few cases reported in the literature so far [17,27–30].

Histopathologically, EBV-MCU is a sharply circumscribed ulcer that is characterised by a polymorphous infiltrate of EBV-positive atypical immunoblasts, which range in size, and other inflammatory cells, such as lymphocytes, plasma cells, eosinophils and histocytes [1]. The atypical immunoblasts show a Hodgkin and Reed–Sternberg (HRS) cell-like morphology, which is difficult to distinguish from those present in classical Hodgkin lymphoma. Presenting a B-cell immunophenotype, these HRS-like cells express CD20, CD30, MUM1, PAX5, OCT-2, CD79a, BOB1 and, in about half of the cases, CD15. A clonal rearrangement of the immunoglobulin genes might be observed, indicating clonal out growths of the EBV-positive B cells. A small rim of surrounding T-lymphocytes can be commonly found. These T cells are mainly CD8-positive and often show a monoclonal or clonal restriction of T-cell receptor (TCR) gene rearrangements, leading to a limited repertoire against EBV epitopes [1]. Furthermore, apoptotic cells and necrosis can be detected as well.

Epstein–Barr virus-encoded small RNA (EBER) in situ hybridization shows an EBV presence in small B cells, plasmacytoid apoptotic cells and immunoblasts [1,13,18].

Especially in aged patients with a natural reduced response of the immune system to new antigens and without any further cause for immunosuppression, EBV-MCU was described as a nearly asymptomatic disease that can regress spontaneously without any treatment [1,13,31–33]. In patients on immunosuppressive drug therapy, complete remission could be successfully observed after a dose reduction of their immunosuppressive medication [8,12,18,19,23–26,34]. However, in some severe and persistent cases, EBV-MCU had to be treated with the monoclonal CD20 antibody rituximab or even with more extended chemotherapy, local radiation therapy, surgical interventions or a combination of these options to receive an adequate response [1,3–7,16–18,35].

## 2. Case Report

We report the case of an 89-year-old male patient who presented with severe odynophagia and left-sided cervical swelling that had been present for more than two weeks. B symptoms, such as weight loss, fever or night sweat were negated. The patient also reported a history of a chronic lymphatic leukaemia (CLL), first diagnosed in 2016. However, the initially started systemic chemotherapy had to be stopped because of severe side effects. There was no evidence of former cigarette smoking or alcohol consumption.

An ear nose and throat (ENT) examination showed painful, indurate, well-circumscribed ulcers on the right palatine tonsil, on the right side of the hypopharyngeal mucosa reaching up to the base of the tongue and on the inside of the lower lip (Figure 1). The blood count showed elevated levels of leukocytes (17.3/nL, reference 4.0–10.0/nL) and c-reactive protein (1.4 mg/dL, reference ≤0.5 mg/dL) and slightly lowered levels of haemoglobin (11.0 g/dL, reference 13.5–17.5 g/dL), erythrocytes (3.8/pL, reference 4.4–5.9/pL) and thrombocytes (131/nL, reference 140–400/nL).

An ultrasound and a computed tomography (CT) of the neck presented bilaterally enlarged lymph nodes with pathological configurations, the largest lymph node measuring 6.5 × 3.0 cm (Figure 2a), as well as an extensive, mostly superficial mucosal defect of the right pharyngeal wall, reaching from the right tonsil and the dorsolateral margin of the tongue up to the level of the glottic region (Figure 2b,c). Beside the bilateral cervical lymphadenopathy, the CT scan of the chest and abdomen showed multiple enlarged and highly suspicious mesenteric and thoracic lymph nodes (Figure 2d). The abdominal ultrasound and CT scan showed an isolated splenomegaly with a spleen size of 16.3 × 9.3 cm (Figure 2e).

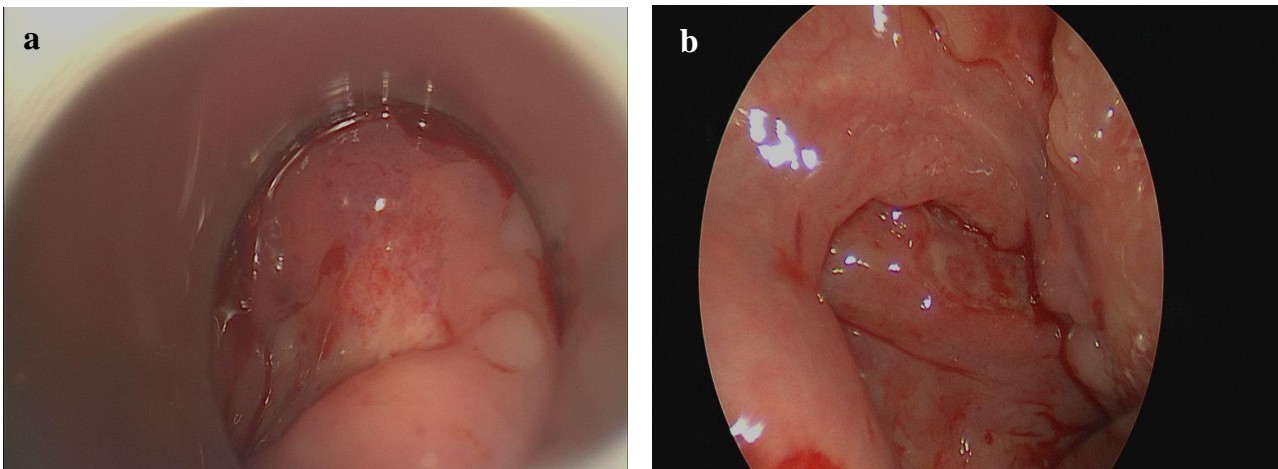

**Figure 1.** Oro- and hypopharyngeal EBV-MCU presenting as a well-circumscribed ulcer on the right side of the hypopharyngeal mucosa (**a**) and on the right palatine tonsil (**b**).

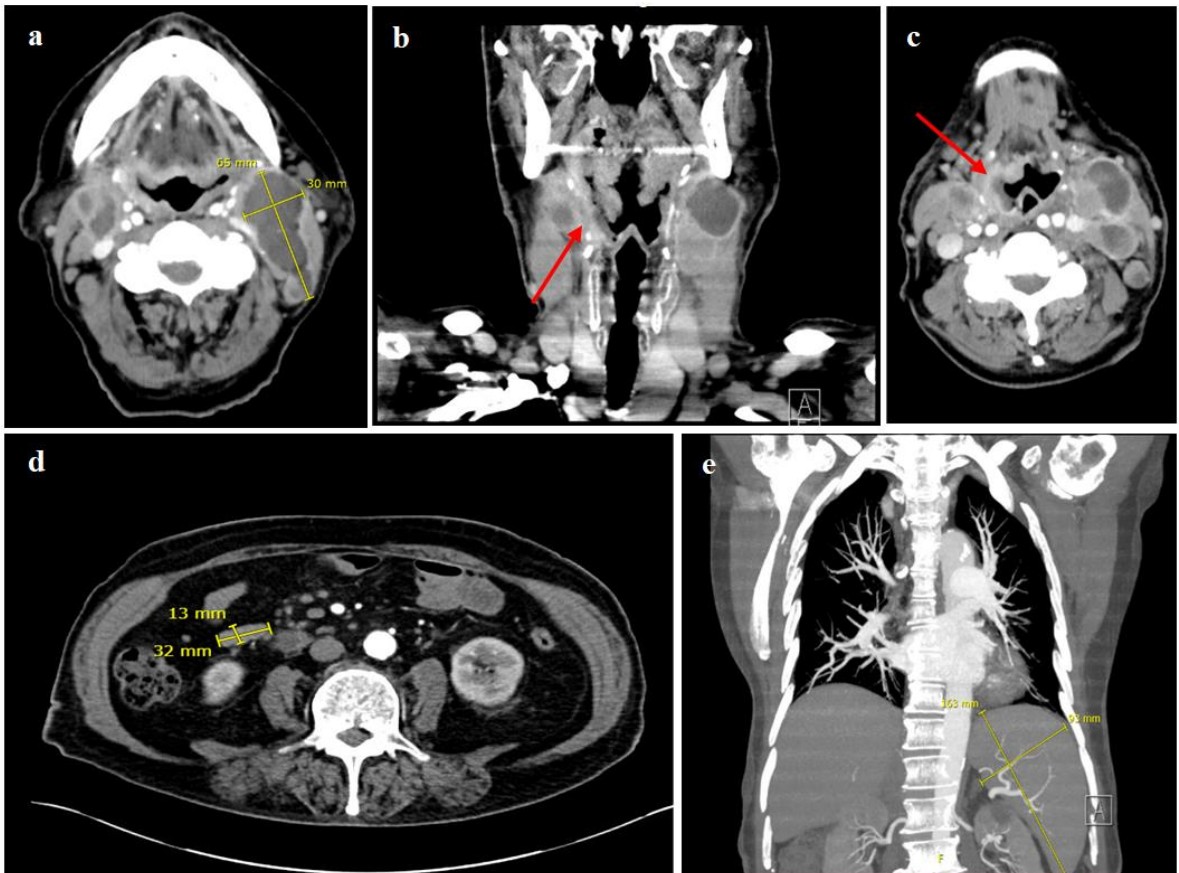

**Figure 2.** CT scan of the neck and chest. (**a**–**c**) showing multiple pathological enlarged cervical lymph nodes on both sides, some of them with central necrosis, the largest measuring 6.5 × 3.0 cm, as well as an extensive mucosal defect with contrast enhancement and partially blurred demarcation reaching from the dorsolateral right side of the tongue, over the right oro- and hypopharynx up to the glottic region, indicating a mucosal ulceration (red arrows). CT scan of the thorax and abdomen (**d**,**e**), demonstrating mesenteric, para-aortal and aortocaval lymph nodes increased in number and size, the largest in the right mid-abdomen measuring 3.2 cm × 1.3 cm (**d**), and a splenomegaly with a spleen size of 16.3 × 9.3 cm (**e**).

An incisional biopsy of the ulcers of the pharynx and the lower lip and a cervical lymph node removal were performed to determine the presence of a squamous cell carcinoma or a lymphoma of the head and neck. All tissue samples were stained with haematoxylin and eosin (H&E), Giemsa and periodic acid-Schiff (PAS) reaction and were immunohisto-chemically tested for a panel of antibodies including CD3, CD20, CD23, CD30 and EBER in situ hybridization. Additionally, a multiplex polymerase chain reaction (PCR) assay in accordance with the standardised protocol of the European BIOMED-2 collaborative study [36] was performed on the samples taken from the lymph node to clonally study rearrangements of the IgH and TCR genes.

The biopsy of the hypopharyngeal ulcer revealed different areas (Figure 3). Below the mucosal erosion, a mixed polymorphous infiltrate of immunoblasts, small lymphatic cells of different sizes and cells with a Hodgkin and Reed–Sternberg (HRS) cell-like morphology (Figure 3a) that showed a significant EBV-activation in the EBER in situ hybridization (Figure 3b) was seen. The pleomorphic large cells showed CD20 and CD30 expression (Figure 3c,d). The proliferation rate in the areas rich in large cells was elevated up to 60% (Ki-67). In addition, next to the ulcerated epithelial areas, focal dense small cell infiltrations could be appreciated (Figure 4). This monotonous infiltrate was CD20-positive with partial co-expression of CD23 (Figure 4a,b).

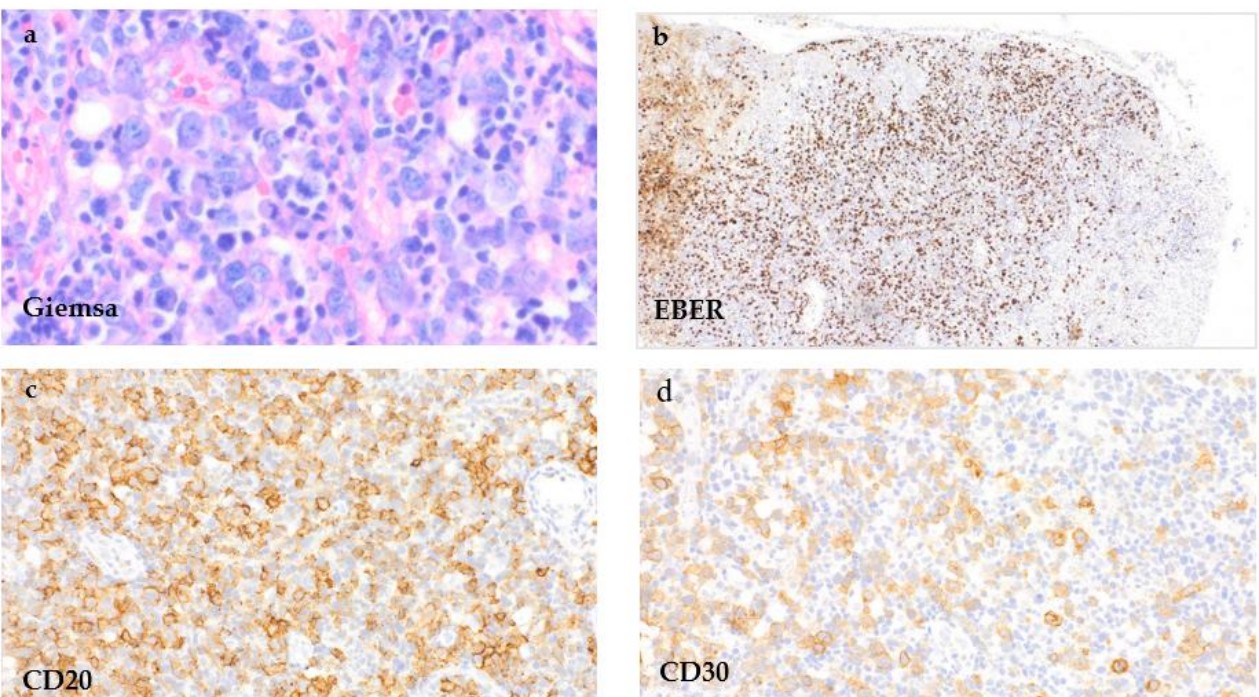

**Figure 3.** EBV-MCU of the hypopharynx. Tissue samples of the hypopharyngeal ulcer taken directly below the epithelial ulceration by incisional biopsy, showing a polymorphous cell infiltrate with HRS-like cells in an inflammatory background ((**a**), Giemsa stain, 400×) that show positivity in the EBER in situ-hybridization ((**b**), EBER, 100×) and for CD20 ((**c**), 400×) and CD30 ((**d**), 400×).

The tissue sample of the lower lip presented a superficial ulcer with a subepithelial polymorphous cell infiltration dominated by small lymphatic B cells and plasma cells. EBV activation in the EBER in situ hybridization could be proven as well. In contrast to the specimen of the hypopharyngeal ulcer, there was no elevated proliferation rate or clear increase in CD30-positive cells. A co-expression of CD23 was slightly noticeable.

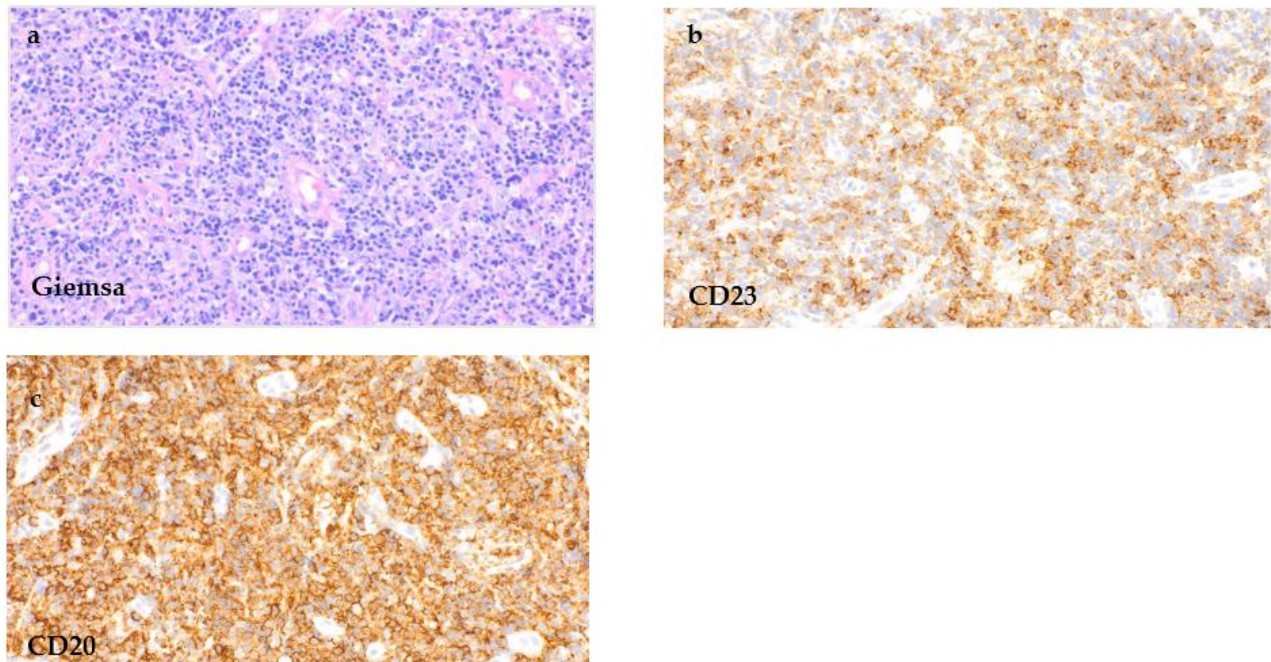

**Figure 4.** CLL/SLL infiltration of the hypopharyngeal ulcer. Tissue samples of the hypopharynx next to the ulcerated area taken by incisional biopsy, showing a significant number of small lymphatic cells ((**a**), Giemsa stain, 100×) with expression for CD20 ((**c**), 400×) and co-expression for CD23 ((**b**), 400×), reflecting the simultaneous infiltration by the known chronic lymphatic leukaemia.

The removed lymph node missed a recognizable basic architecture with a pseudo-follicular pattern of growth and was mainly infiltrated by small lymphocytes, rounded lymphocyte-like cells with a lump-like chromatin and some immunoblasts (Figure 5a). The proliferation rate was 10–30% (Ki-67). An immunohistochemical analysis remarked a strong expression of CD20 (Figure 5c) as well as CD79a (Figure 5f) beside a further expression of CD3. Only single EBV-positive bystander small lymphocytes with reactivity were seen in the EBER in situ hybridization (Figure 5g). Furthermore, the small lymphatic cells of the dense B-cell region showed a co-expression of CD5 and CD23 (Figure 5b,d), leading altogether to the phenotype of a chronic lymphatic leukaemia (CLL)/a small lymphocytic lymphoma (SLL). Multiplex PCR revealed monoclonal immunoglobulin heavy chain (IgH) gene rearrangements. The histologic pattern of the lymph node tissue, in combination with the immunohistochemical findings, presented the manifestation of a CLL/SLL. The analysis of the ulcers of the hypopharynx and lower lip matched with the diagnosis of an EBV-MCU and showed a simultaneous infiltration of the CLL/SLL as well.

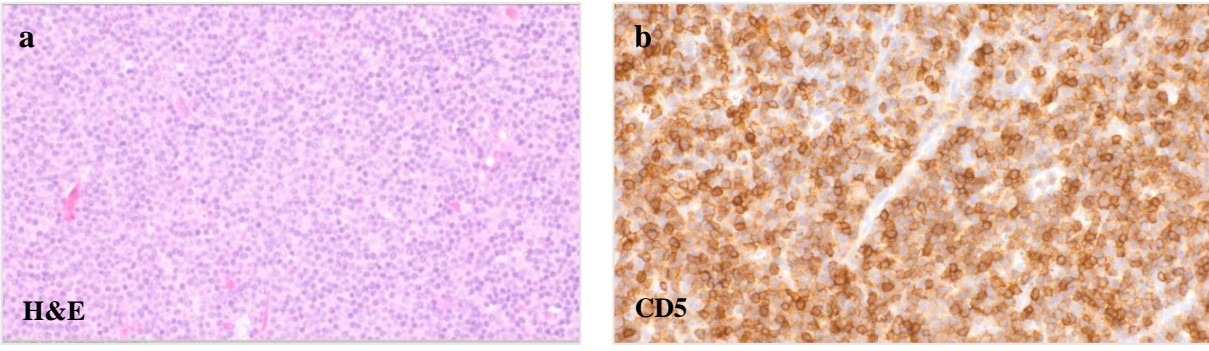

**Figure 5.** *Cont.*

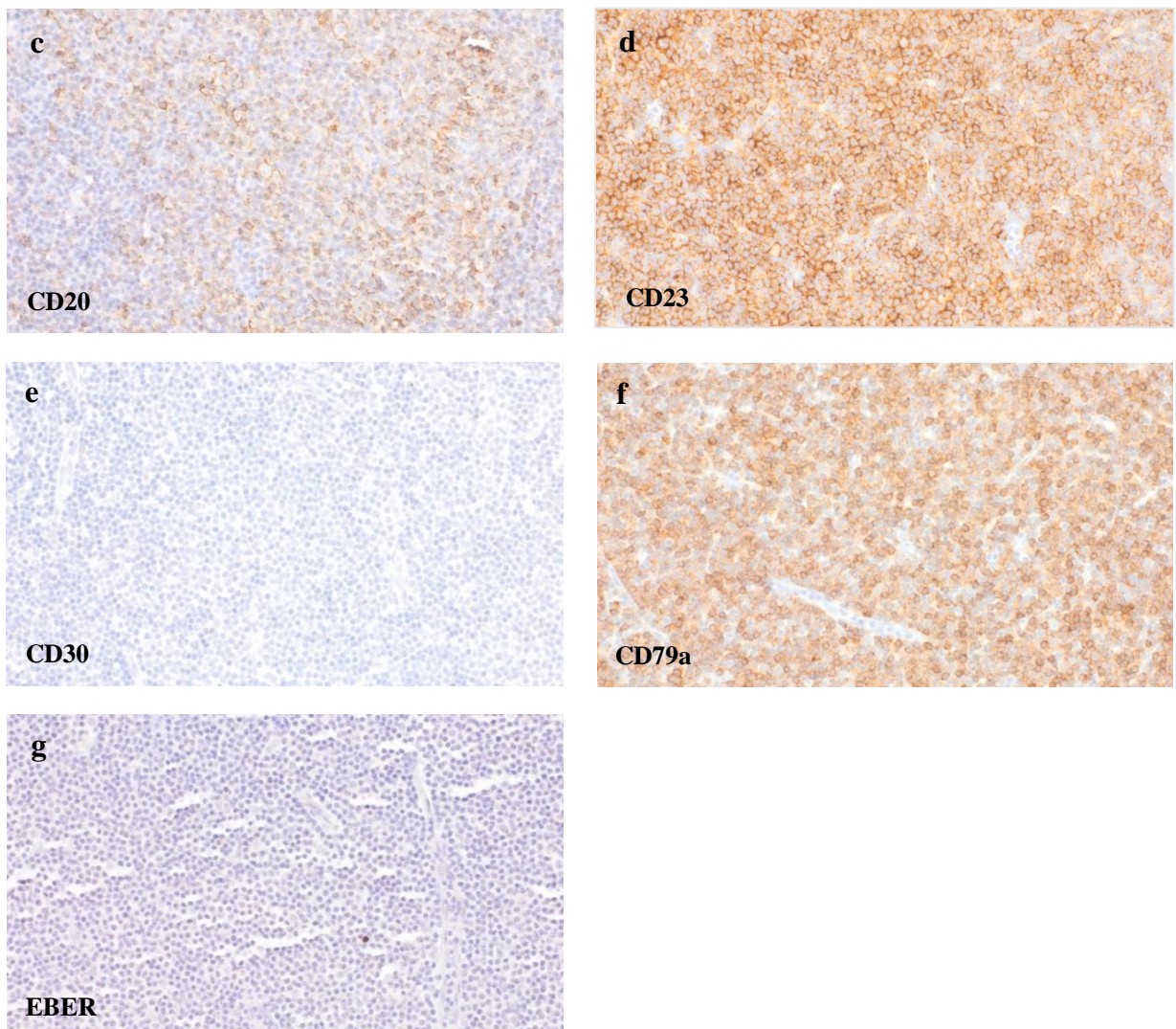

**Figure 5.** CLL/SLL infiltration of the lymph node. Tissue samples of the removed cervical lymph node, showing a dense cell infiltrate of small lymphocytes ((**a**), H&E staining, 400×) with a co-expression of CD5 ((**b**), 400×), CD20 ((**c**), 400×), CD23 ((**d**), 400×) and CD79a ((**f**), 400×). CD30 ((**e**), 400×) remained negative, and only single small EBV-positive bystander cells were seen in EBER in situ hybridization ((**g**), 400×).

Chemotherapy was planned as the definitive treatment of the relapsing CLL/SLL. Unfortunately, our patient died of an unknown reason two weeks after being discharged from the hospital without therapy being started. Thus, further follow up during and after treatment was not available.

### 3. Material and Methods

The PubMed database was searched from 1 January 1993 to 8 April 2022 with the key words "Epstein Barr virus", "EBVMCU", "mucocutaneous ulcer", "head and neck ulcer" and "EBVMCU head and neck" to identify all EBV-MCU cases of the head and neck region published so far. All cases published about an EBV MCU in the indicated period were included into the following review without exclusion criteria. A total of 181 cases were identified.

Information on the underlying source of immunosuppression, EBV-MCU localisation, therapy and response to therapy had been collected and summarised in Table 1.

## 4. Results

Table 1 gives an overview of all EBV-MCU cases of the head and neck published so far. In total, 181 cases were recorded between January 2001 and December 2022.

Contrary to all the other identified cases, in our case, histopathology confirmed a simultaneous manifestation of an EBV-MCU and another malignant disorder within the same lesion, which had not been described in the literature before.

A total of 76% ($n$ = 137) of all cases were caused by immunosuppressive drug therapy, mainly attributed to methotrexate in rheumatoid arthritis (64%, $n$ = 88), followed by age-associated immunosenescence in 17% ($n$ = 30) and less common causes, including primary immunodeficiencies (2%, $n$ = 4) and HIV (2%, $n$ = 3). The ulcers were localised on the oral mucosa in 56% ($n$ = 102), on the (oro)pharyngeal mucosa in 23% ($n$ = 42), on the skin in 8% ($n$ = 15), on the nasal mucosa in 2% ($n$ = 4) and on the oesophageal mucosa in 2% ($n$ = 3). EBV-MCU of the oral cavity and the oropharynx predominantly affects the gingiva, the palate, the buccal mucosa, the tonsils and the base of the tongue. Our patient presented a one-sided EBV-MCU that was mainly located on the oro- and hypopharyngeal mucosal wall, with an extension from the right palatine tonsil and the base of the tongue up to the hypopharyngeal level of the glottic space. Particularly, the ulcers of the oral cavity and the oropharyngeal mucosa can show a wide expansion with extensive destruction of soft tissue, bone erosion and tooth loss leading to mutilation and functional restrictions [3,5,37,38]. So far, there are only two reported cases of an EBV-MCU affecting the deeper parts of the pharynx and the larynx [38].

EBV-MCU was often described as a self-limiting disease that does not require any therapy [1,13,31,33,39]. Patients with EBV-MCU resulting from age-associated immunosenescence were mentioned to respond well to conservative treatment, showing spontaneous regression of the ulcers. However, only 11 out of 181 reviewed cases (6%) ultimately showed a spontaneous remission at all (Table 1) [1,9,13,31–33]. Complete remission was observed in 68% of all patients ($n$ = 123) and was mainly attributed to a reduction of the immunosuppressive therapy alone (72%, $n$ = 88 of 123 patients with complete remission). However, there were also reported cases of persistent and more dynamic forms of EBV-MCU that had to be actively treated. Systemic monoclonal antibody therapy, local radiation, chemotherapy, surgical excision or a combination of these treatment modalities were observed to be successful in most cases (69%, $n$ = 34 out of 49 treated patients). EBV-MCU with a sustained disease response responded sufficiently to additional R-CHOP (rituximab, cyclophosphamide, doxorubicin, vincristine and prednisone) chemotherapy (26%, $n$ = 13 of 49 treated patients), radiotherapy (10%, $n$ = 5 of 49 treated patients) or a combination of both (4%, $n$ = 2 of 49 treated patients), all reaching complete remission after therapy.

Administering the monoclonal anti-CD20 antibody rituximab in weekly dosages of 375 mg/m$^3$ offered an additional therapeutic benefit in several reports [3,4,7,27,37,40,41]. A surgical excision of localised ulcers was mainly performed to exclude a carcinoma or another malignancy [7,17,42] or was necessary to reconstruct tissue defects after regression [15]. The combination of surgical resection and reduced immunosuppression reached complete remission in all cases (12%, $n$ = 6 of 49 treated patients). Only four cases (2%, $n$ = 4 of 181 reviewed cases) presented a relapsing and remitting course.

There are few new therapeutic targets in EBV-MCU treatment being reported. Kleinman et al. treated a 61-year-old woman with the diagnosis of a hypogammaglobulinemia with three cycles of brentuximab (1.8 mg/kg, every 3 weeks) and an anti-CD30 monoclonal antibody and achieved partial remission of an oesophageal EBV-MCU after a failed combined therapy with rituximab (375 mg/m$^3$, four weekly doses) and monthly immunoglobulin infusions (600 mg/kg) [29]. Another potential therapeutic approach in severe and persistent EBV-MCU could be autologous or allogeneic infusions of EBV-specific cytotoxic T cells (EBV-CTLs) [3,29,37]. Khazal et al. reported on a 75-year-old woman with a persistent, gingival infiltrating EBV-MCU that did not respond to four weekly doses of rituximab (375 mg/m$^3$) but regressed completely after two cycles of EBV-CTL infusions ($2 \times 10^6$/kg cell dose per infusion, three weekly infusions, 2 weeks off per cycle) without

showing recurrence within 12 months after therapy [37]. Xu et al. present a case of a 76-year-old woman treated with interferon-alpha1b for an EBV MCU in the oral cavity. Complete remission could be achieved in four weeks after an increase in dose from 30 to 50 μg three times a week to 100 μg three times a week [43].

**Table 1.** Overview of EBV-MCU cases of the head and neck, including source of immunosuppression, localization, treatment and outcome [1,3–8,11,12,14,16–19,21,23–25,27–30,32,34,35,37–39,44–56].

| Author/Reference | Number of Cases | Source of Immunosuppression | Localisation | Treatment and Response |
|---|---|---|---|---|
| Alamoudi et al., 2022 [34] | 1 | primary immunodeficiency + EBV-negative follicular lymphoma | oral cavity | unknown |
| Aldridge et al., 2017 [44] | 1 | RA (MTX, prednisolone) | oral cavity | reduced IS; CR |
| Attard et al., 2012 [23] | 1 | RA (MTX) | oral cavity/tongue | reduced IS; CR |
| Au et al., 2011 [27] | 1 | primary immunodeficiency | oral cavity/gingiva | RTX; CR |
| Au et al., 2016 [4] | 1 | Crohn's disease (AZA) | oropharynx/base of the tongue | RTX, acyclovir; CR |
| Bunn et al., 2015 [28] | 2 | HIV | oral cavity | unknown |
| Chen et al., 2017 [31] | 1 | RA (MTX) | oral cavity/gingiva | SR |
| Chen et al., 2019 [12] | 1 | RA (MTX, prednisolone) | oral cavity | reduced IS; CR |
| Daroontum et al., 2019 [13] | 1 | senescence | oral cavity/gingiva | SR |
| Dojcinov et al., 2010 [1] | 20 | senescence (15) RA (MTX) (3) SLE (CYA) (1) sarcoidosis, myasthenia gravis (AZA) (1) | oral cavity (11) oropharynx/tonsils (5) skin/lip (3) oesophagus (1) | unknown (4) SR (4), RD (3), SD (1) reduced IS; CR (3) radiotherapy; CR (3) chemotherapy (R-CHOP) ± radiotherapy; CR (2) |
| Donzel et al., 2021 [46] | 1 | RA (MTX) | skin/scalp | reduced IS, PR |
| Eleftheriadis et al., 2021 [47] | 1 | solid organ transplantation (methylprednisolone, mycophenolate) | oral cavity | RTX + haemodialysis + surgery + antibiotics, CR |
| Forster et al., 2022 [48] | 1 | multiple myeloma (stem cell transplant, lenalidomide) | oropharynx/tonsil | reduced IS, CR |
| Fujimoto et al., 2021 [49] | 1 | RA (MTX) | oral cavity | SR |
| Hart et al., 2014 [7] | 5 | solid organ transplantation | lip, oral cavity (4) oesophagus (1) | reduced IS; CR (3) reduced IS + surgery; CR (1) reduced IS + RTX; CR (1) |
| Hashizume et al., 2012 [24] | 1 | polymyositis (MTX) | skin/lip, nose, eyelid | reduced IS; CR |
| Hujoel et al., 2018 [57] | 1 | Crohn's disease (AZA) | oral cavity/palate | topical triamcinolone; PD, reduced IS; CR |
| Ikeda et al., 2020 [8] | 32 | RA (MTX ± tacrolimus)(28) senescence (3) polycythaemia (hydroxycarbamide) (1) | oral cavity (19) oropharynx/tonsils (7) pharynx (3) nasal cavity (2) skin/angle of mouth (1) | reduced IS; CR (16), PR (7) chemotherapy (R-THPCOP/R-CHOP) ± reduced IS; CR (3) unknown (6) |
| Kanemitsu et al., 2015 [15] | 1 | SLE nephritis (mycophenolate) | oral cavity/gingiva | unknown |
| Kawamura et al., 2022 [50] | 2 | unknown (2) | oral cavity/gingiva (1) oropharynx/tonsil (1) | unknown (2) |

**Table 1.** *Cont.*

| Author/Reference | Number of Cases | Source of Immunosuppression | Localisation | Treatment and Response |
|---|---|---|---|---|
| Khazal et al., 2020 [37] | 1 | CLL and Merkel cell carcinoma (chemotherapy, radiotherapy, pembrolizumab) | oral cavity/gingiva | RTX; allogenic EBV-CTL infusions; CR |
| Kleinmann et al., 2014 [29] | 1 | hypogammaglobulinemia | oesophagus | RTX + immunoglobulin infusion; SD, brentuximab; PR |
| Kunmongkolwut et al., 2022 [51] | 1 | RA (MTX + leflunomide) | oral cavity/ palate + gingiva | reduced IS, CR |
| Li et al., 2020 [5] | 1 | pemphigus vulgaris (mycophenolate, prednisolone) | oral cavity/gingiva | reduced IS + surgery (bone sequestrectomy); CR |
| Magalhaes et al., 2015 [32] | 1 | senescence | oral cavity/palate | SR |
| Masuoka et al., 2022 [53] | 1 | RA (MTX) | hypopharynx | unknown |
| McCormack et al., 2018 [58] | 1 | senescence | oral cavity/gingiva | unknown |
| McGinness et al., 2012 [25] | 1 | pemphigoid (AZA, prednisolone) | oral cavity/buccal mucosa | reduced IS; CR |
| Moriya et al., 2020 [16] | 1 | RA (MTX, prednisolone), DLBCL | pharynx | chemotherapy (R-CHOP); CR |
| Nakauyaca et al., 2016 [38] | 1 | RA (MTX) | pharynx | death (bleeding, heart failure) |
| Natkunam et al., 2017 [17] | 9 | immunosuppression, not further specified (5) unknown (3) CHARGE syndrome (1) | oral cavity (6) nasopharynx (1) nasal mucosa (1) skin/lip (1) | unknown (3) surgery ± reduced IS; CR (2) RTX; RD (1) reduced IS; CR (1) RTX + chemotherapy (R-CHOP) + reduced IS; CR (1) chemotherapy + HCT; CR, RD and death (1) |
| Nelson et al., 2016 [21] | 1 | myelodysplastic syndrome, HCT (fludarabine, busulfan, MTX, tacrolimus) | oral cavity/buccal mucosa | reduced IS; CR |
| Obata et al., 2021 [54] | 10 | RA (MTX ± prednisolone/tacrolimus/ bucillamine) (10) | oral cavity/gingiva (10) | reduced IS, CR (10) |
| Ohata et al., 2017 [55] | 14 | senescence (5) RA (MTX) (9) | oral cavity/gingiva, tongue, buccal mucosa | reduced IS; CR (7) chemotherapy (R-CHOP/R-COEP); CR (4) unknown (3) |
| Pina-Oviedo et al., 2018 [56] | 1 | CLL (fludarabine, cyclophosphamide, RTX, bendamustine) | oropharynx/base of the tongue | unknown/death (EBV + DLBCL) |
| Prieto-Torres et al., 2019 [9] | 5 | HIV (1) RA (MTX, prednisolone) (2) senescence (2) | oral cavity (3) pharynx (1) skin/forehead (1) | reduced IS; CR (2) chemotherapy (R-CHOP); CR (1) radiotherapy; CR (1) SR (1) |
| Ravi et al., 2018 [59] | 1 | RA (MTX) | oral cavity/buccal mucosa | reduced IS, metronidazole; CR |

**Table 1.** *Cont.*

| Author/Reference | Number of Cases | Source of Immunosuppression | Localisation | Treatment and Response |
|---|---|---|---|---|
| Ritter et al., 2020 [30] | 1 | SLE (MTX, AZA, hydroxychloroquine, prednisone) | skin/forehead | reduced IS; PR |
| Roberts et al., 2016 [3] | 1 | unknown | oral cavity | RTX; PR/PD, subsequent radiotherapy; CR |
| Sadasivam et al., 2014 [26] | 1 | RA (MTX) | oral cavity/palate | reduced IS; CR |
| Sadiku et al., 2012 [35] | 1 | senescence | skin/neck | chemotherapy (R-CHOP); CR |
| Satou et al., 2017 [22] | 1 | DLBCL, chemotherapy + HCT | oral cavity/tongue | SR; death (EBV + PTLD) |
| Satou et al., 2019 [18] | 9 | RA (MTX ± infliximab/etanercept/ tacrolimus/adalimumab) | oropharynx (7) oral cavity/gingiva, tongue (2) | reduced IS; CR (7) chemotherapy (R-CHOP, ABVD); CR (2) |
| Shiraiwa et al., 2020 [19] | 15 | RA/Sjögren's disease (MTX) | oral cavity (5) pharynx (3) oropharynx/tonsils (3) skin (3) nasal septum (1) | reduced IS; CR (11) unknown (4) |
| Sinit et al., 2019 [60] | 1 | recurrent sinopulmonary infections (prednisolone) | nasopharynx | radiotherapy; CR; DLBCL |
| Takada et al., 2021 [61] | 15 | RA (MTX) (15) | unknown (15) | reduced IS, CR (13) unknown (2) |
| Teixeira et al., 2018 [42] | 1 | Crohn's disease (AZA, infliximab), WM/LPL | oropharynx/tonsil | surgery; CR |
| Tsusumi et al., 2020 [40] | 1 | senescence | oropharynx | RTX; CR |
| Uygun et al., 2022 [62] | 1 | severe aplastic anaemia (stem cell transplantation, CYA) | oropharynx | methylprednisolone + sirolimus, CR |
| Vatsayan et al., 2017 [36] | 1 | T-ALL (nelarabine/Capizzi, mercaptopurine, MTX) | oropharynx/tonsil | reduced IS + RTX; CR |
| Wanderlei et al., 2019 [41] | 1 | Sjögren's dieseade (MTX) | oral cavity | SR |
| Xu et al., 2021 [43] | 1 | unknown | oral cavity/buccal mucosa | interferon alpha-1b, CR |
| Yamakawa et al., 2014 [20] | 3 | RA (MTX) | skin/eyelid (2) oral cavity/buccal mucosa (1) | reduced IS; CR (2); CR and death (myelitis, sepsis) (1) |
| **In Total** | **181** | **iatrogenic/immunosuppressive therapy: 137 cases age-associated immunosenescence: 30 cases unknown: 7 cases primary immunodeficiency: 4 cases HIV: 3 cases** | **oral cavity: 102 cases oropharynx: 30 cases skin: 15 cases pharynx: 12 cases nasal mucosa: 4 cases oesophagus: 3 cases** | **complete remission: 123 cases partial remission: 10 cases spontaneous remission: 11 cases recurrent disease: 4 cases stable disease, no treatment: 1 case unknown: 32 cases** |

ABVD = adriamycin, bleomycin, vinblastine, dacarbazine; AZA = azathioprine; CYA = cyclosporine A; HCT = haematopoietic cell transplantation; IS = immunosuppression; MTX = methotrexate; RTX = rituximab; R-CHOP = rituximab, cyclophosphamide, vincristine, adriamycin, prednisolone; CR = complete remission; PD = progressive disease; PR = partial remission; RD = recurrent disease; SD = stable disease; SR = spontaneous remission; CTL = cytotoxic T lymphocytes; CLL = chronic lymphatic leukaemia; DLBCL = diffuse large B-cell lymphoma; LPL = lymphoplasmacytic lymphoma; RA = rheumatoid arthritis; SLE = systemic lupus erythematosus; T-ALL = T-cell acute lymphoblastic leukaemia; WM = Waldenström macroglobulinae.

The literature review exposed several cases of non-disease-associated deaths after the diagnosis of EBV-MCU: two patients developed a more malignant EBV-positive lymphoproliferative disease (EBV + DLBCL and EBV + PTLD) and died [22,56], an 87-year-old woman died of congestive heart failure [9] and a 67-year-old male patient had fatal bleeding after a tonsillectomy with cardiopulmonary arrest [38]. However, there had been two further cases described where it is not clear whether death can be attributed to EBV-MCU. The first case of a 16-year-old male patient with a primary immunodeficiency due to a CHARGE syndrome (coloboma, heart defects, atresia of the nasal choanae, retardation of growth, genital abnormalities and ear abnormalities/deafness) was presented in a workshop report in 2017 [17]. He presented with an EBV-MCU of the nasopharyngeal mucosa and a lymph node swelling that successfully regressed completely after chemotherapy and a haematopoietic stem cell transplantation. In the follow-up, the disease had relapsed and the patient died. The other case reported a 64-year-old female patient with rheumatoid arthritis who suffered under a methotrexate-associated EBV-MCU of the buccal mucosa that showed a complete response after the withdrawal of the immunosuppressive therapy [20]. After one month she died from myelitis and sepsis.

In both cases, the authors did not give any further information and, thus, the circumstances leading to death remain unclear.

## 5. Discussion

EBV-associated LPDs have gathered attention in recent years. The WHO classified these diseases into four groups based on the clinical circumstances they arise: LPDs with primary immune disorders, lymphomas associated with HIV infections, post-transplant LPDs and other iatrogenic immunodeficiency-associated LPDs.

In 2017, the EBV-positive mucocutaneous ulcer (EBV-MCU) was classified as a rare new entity of the cutaneous lymphoproliferative B-cell diseases by the WHO that primarily affects the oral and oropharyngeal mucosa as a result of an EBV reactivation due to immunosuppression. Despite most of the described cases, EBV-MCU can be a painful and debilitating disease that does not always regress spontaneously and needs further treatment. The histologic and phenotypic characteristics with a presence of Hodgkin and Reed–Sternberg (HRS)-like cells make it challenge to distinguish it from other lymphoma types.

We present a case of an 89-year-old man with multifocal EBV-MCU of the head and neck that was simultaneously infiltrated by a recurrent CLL/SLL.

### 5.1. EBV-MCU and Haematologic Malignancies

EBV-MCU usually appears because of a medically induced immunosuppression or because of advanced-age-associated immunosenescence. Beside the age, which is seen as a co-factor, immunodeficiency in our patient was mainly caused by a relapsing chronic lymphatic leukaemia. CLL was mentioned to be one of the most common haematologic malignancies known to produce immune defects independent of therapy [56]. The histopathologic and immunohistochemical features of the ulcers of the pharynx and the lower lip showed the characteristic pattern of an EBV-MCU with a B-cell polymorphous infiltrate, including HRS-like cells and an expression for CD20 and CD30, and also showed areas consistent with simultaneous CLL/SLL infiltration. Thus, the manifestation of the CLL/SLL, clearly presented in the lymph node biopsy, was also detectable in both ulcers. The simultaneous manifestation of an EBV-MCU and another malignant disorder within the same lesion had not been described in the literature before.

There are few described cases of a synchronous or metachronous appearance of EBV-MCU and different lymphoproliferative diseases [13,16,22,56], with only three cases occurring in association with a present haematologic malignancy [37,41,56].

Khazal et al. reported on a 75-year-old male patient with a medical history of CLL many years earlier that had been treated with rituximab, fludarabine and cyclophosphamide as well as recurrent left arm Merkel cell carcinoma that was treated by radiation

therapy, surgery and pembrolizumab [37]. He presented with a painful gingival ulcer with progress to mandibular infiltration that fit the histological characteristics of an EBV-MCU. A bone marrow evaluation showed 60–70% CLL infiltration. Although the large HRS-like cells within the ulcer biopsy raised the possibility of a Hodgkin-type Richter transformation of the CLL, the diagnosis was finally based on clinical presentation and course.

The second case described a 29-year-old adult with intermediate-risk acute T-cell lymphoblastic leukaemia (T-ALL) who was treated with Children's Oncology Group proto-col AALL0434 chemotherapy, including methotrexate and mercaptopurine, at the time of left-sided throat pain caused by an exudative left palatine tonsillar ulcer [41]. The biopsy of the ulcer showed a polymorphous infiltrate of lymphocytes, neutrophils, plasma cells and histiocytes as well as HRS-like cells. These cells were positive for CD20, CD30, CD45 and EBER and were negative for CD5, CD10 and CD15, which was, all in all, consistent with the diagnosis of an EBV-MCU. As in the previous case, there were no simultaneous histological signs of the T-ALL within the ulcer. After the EBV-MCU did not regress despite a reduction of the immunosuppressive drug therapy, a single dose of rituximab (375 mg/m$^2$) was administered and the ulcer resolved within two weeks.

Pina-Oviedo et al. analysed the clinical and pathological features of patients who de-veloped a lymphoproliferative disorder after a therapy for a haematologic malignancy [32]. One of these patients, an 82-year-old woman with CLL under treatment with fludarabine, cyclophosphamide and rituximab, developed an EBV-MCU at the base of the tongue. Histopathology and immunohistochemistry were typical for EBV-MCU, with cells positive for CD20, CD30, CD45/LCA, CD79a, PAX5, OCT2 and EBER and negative for CD3, CD15 and BOB1. Thus, unlike our case, no infiltration by the CLL was noted. Three months later, she developed cervical lymph nodes that were infiltrated by an EBV-positive diffuse large B-cell lymphoma (DLBCL) and died. While DLBCL was the most common type of LPD after cancer therapy, the primary haematologic malignancy among this study group was predominantly CLL, assuming a certain predisposition for developing an EBV-positive lymphoproliferative disorder (EBV + LPD).

## 5.2. EBV-MCU and Other LPDs, a Challenge of Differentiation

EBV-MCU shares morphologic and phenotypic characteristics with more aggressive forms of lymphoproliferative disorders, such as EBV-positive diffuse large B-cell lym-phoma (DLBCL), classic Hodgkin lymphoma (cHL), post-transplant lymphoproliferative disorder (PTLD), plasmablastic lymphoma (PBL), anaplastic large cell lymphoma (ALCL) or lymphomatoid granulomatosis (LyG), which sometimes makes it extremely difficult to distinguish.

## 5.3. EBV-Positive Diffuse Large B-Cell Lymphoma (DLBCL)

EBV-positive diffuse large B-cell lymphoma (DLBCL) is a high-grade lymphoma with a poor outcome. It presents a wide range of morphological features, from polymorphous infiltrates of HRS-like cells, immunoblasts and plasma cells in an inflammatory background of histiocytes and lymphocytes with areas of prominent necrosis and angioinvasion to a T-cell-like or more monomorphic pattern [39]. The cells are positive for CD20, CD19, CD79a, PAX5, OCT2, BOB1, MUM1 and CD30, with variable co-expression of CD15. CD10 is mostly negative and CD45 and BCL6 show a variable expression. There is a widespread positivity for EBER. These almost indistinguishable similarities to EBV-MCU, cHL may be misleading and all clinical, histologic, and immunohistochemical features must be considered to avoid misdiagnosis [14]. Both diseases predominantly occur in elderly people with presumed immunosenescence, but while EBV-MCU shows a localised nature and often a self-limiting course, DLBCL occurs as a generalised, extranodal progressive disease. The sharply circumscribed EBV-MCU with a small rim of T cells at the base can be differentiated from the more infiltrative pattern in DLBCL. Considering the findings of Ohata et al., a mutational panel and the absence of CD10 and BCL6 expression in EBV-MCU can be supportive [55].

Furthermore, Satou et al. observed that, in contrast to DLBCL and cHL, immune evasion via the programmed cell death protein 1/programmed cell death 1 ligand 1 (PD1/PD-L1) pathway is usually absent in EBV-MCU [18]. However, Daroontum et al. reported a case of a PD-L1-positive EBV-MCU in a patient with multiple EBV-driven lymphoproliferative B-cell disorders [45]. All lesions had PD-L1 expression in the EBV-positive large B cells and HRS-like cells, but there was evidence for a clonal relationship among these lesions by PCR analysis for IGH. The majority of DLBCL show clonal IGH and restricted or oligoclonal TCR rearrangements because of a reduced T-cell repertoire [1]. EBV DNA serum levels often correlate with the burden of disease [39].

The clinical presentation of EBV-MCU, with its strictly localised occurrence and the absence of mass lesions, is crucial in distinguishing these two entities [10,14,16]. There are some reports of lesions that were initially diagnosed as DLBCL before being reclassified as EBV-MCU [9], and, therefore, special attention should be drawn to their similarity.

### 5.4. Classic Hodgkin Lymphoma (cHL)

Classic Hodgkin lymphoma (cHL) is one of the most common lymphoma types worldwide. It is associated with a non-obligatory presence of EBV infection and most commonly leads to an impressive cervical and mediastinal lymphadenopathy. Patients are often presenting B symptoms. cHL shares some (immune)histological characteristics with EBV-MCU, especially the presence of CD30-positive HRS cells co-expressing CD15 in a strong inflammatory background. These cells are also positive for MUM1. The tumour cells show a downregulation of B-cell markers and transcription factors such as OCT2 and BOB1, with a remaining slight expression of PAX5, whereas EBV MCU more often expresses CD 20 [10,14].

In EBV-associated cHL, typically only the HRS cells are positive in EBER in situ hybridization, which is an important fact regarding the differentiation to EBV-MCU in which EBER positivity can be detected in several different cell types. In addition, EBV MCU displays an EBV latency type III with expression of the Epstein–Barr virus nuclear antigen 2 (EBNA 2), whereas cHL presents latency type 0/I or II and is negative for EBNA 2. At least HRS cells express PD-L1 ligands and consistently express LMP1 that starts the NF-kappa B and JAK/STAT signalling pathway, leading to proliferation [10,14]. Applying to this entity as well as for the DLBCL, the clinical presentation of the EBV MCU with strict localised and superficial occurrence is crucial in differentiating these two entities [10].

### 5.5. Plasmablastic Lymphoma (PBL)

Another important EBV-associated LPD that also mainly occurs in the oral cavity is the plasmablastic lymphoma (PBL), a highly aggressive form of non-Hodgkin lymphoma (NHL) with an immunophenotype of terminally differentiated B cells and a loss of typical B-cell antigen expression. The large plasmablasts lack CD20, CD19, CD45 and PAX5 and show an expression of plasma cell markers that are positive for CD79a, MUM1, CD38 and CD138 [12,14]. The cells show an overall expression of MYC protein, and the proliferation rate is high (Ki-67). Most cases are EBV-positive in the EBER in situ hybridization. The PBL is associated with immunosuppression, mainly due to HIV infections, and its localisation in the head and neck region makes it important as a potential differential diagnosis of EBV-MCU. Beside the immunohistochemical differences, PBL shows a monomorphous pattern with sheets of plasmablasts, whereas the EBV-MCU is characterised by a typical polymorphous pattern [9,12]. In contrast to the benign course of EBV-MCU, the prognosis of PBL is poor, with high observed mortality rates.

### 5.6. Post-Transplant Lymphoproliferative Disorder (PTLD)

Post-transplant lymphoproliferative disorder (PTLD) is an immunodeficiency-related proliferation of B cells that are latently infected with EBV after solid organ transplantation and is another important disease to consider in the differential diagnosis of EBV-MCU [7]. The symptoms are nonspecific and can be like those in infectious mononucleosis. PTLD

can form tumour masses that may obstruct organs or spread further, leading to organ dysfunctions. The monoclonal form has an especially aggressive course. The histological features contain the full range of B-cell maturation, from immunoblasts to plasma cells and variable-sized lymphocytes. The HRS-like cells are positive for CD20 and CD30, while CD15 has been reported to be negative. There is an expression of LMP1, inducing an uncontrolled cell proliferation. EBER is mainly positive for EBV. Blood levels of EBV DNA measured by PCR have been shown to predict the development of PTLD [31]. Hart et al. detected EBV DNA in the blood of patients with solid organ transplantation in about 80% of all cases, presenting a quantifiable EBV viremia [7,31]. This viremia usually does not present in patients with EBV-MCU and, therefore, can be a further distinguishing feature [17,21,59]. Nevertheless, the presence of EBV DNA in the blood does not automatically exclude the diagnosis of EBV-MCU [9]. EBV DNA and antibody levels in the blood of patients with EBV-MCU has not been regularly measured and should be further investigated. EBV MCU can arise with typical clinical presentation and corresponding histomorphology in the background of immunodeficiency caused within the post-transplant setting. In those cases, no formal differential diagnosis is necessary, but the lesion should be reported as polymorphous PTLD with features of an EBV-MCU [10].

### 5.7. Anaplastic Large Cell Lymphoma (ALCL)

Anaplastic large cell lymphoma (ALCL) is a type of NHL with the presence of large pleomorphic cells that are positive for CD30 and is another differential diagnosis of EBV-MCU due to histological similarities. It typically presents at a late stage with systemic symptoms as a general lymphadenopathy, extranodal disease or as a cutaneous ulcer. A rare subtype can occur after breast implant reconstruction. The cells are negative for CD3 and for EBV in the EBER in situ hybridization [17,63].

### 5.8. Lymphomatoid Granulomatosis (LyG)

Lymphomatoid granulomatosis (LyG) is an EBV-driven B-cell LPD that is associated with immunosuppression with angiodestructive features and is characterised by a cell infiltrate of small T cells and variable numbers of large atypical immunoblasts and HRS-like cells that are positive for CD20, CD30, LMP1 and EBER but not usually for CD15. The differentiation to EBV-MCU merely relies in the clinical presentation with lung lesions or affliction of the central nervous system [33].

### 5.9. Head and Neck EBV-MCU Management

Beside all these histological similarities to other lymphoproliferative disorders, EBV-MCU predominantly occurs in the mucosa of the oral cavity and the oropharynx and, therefore, mimics squamous cell carcinomas at first sight. As these two entities can clinically present in a similar morphology and the treatment of each is completely different, it is very important to take an early biopsy to exclude an oral or oropharyngeal squamous cell carcinoma. Of course, this also applies to ulcerative lesions of the skin in the head and neck region, as with the ulcer of the lower lip in our patient.

As EBV-MCU is still a rare entity of lymphoproliferative disorders, it is important to mind it as a potential disease in elderly and immunocompromised patients with persistent cutaneous or mucosal ulcerative lesions. An early biopsy must be taken to distinguish it from other malignant disorders. Imaging, a blood examination, including EBV serology, and, in some uncertain cases, a bone marrow biopsy can be helpful to exclude a systemic lymphoproliferative disorder.

Since the awareness for EBV-MCU is currently still low and ulcerative lesions can be easily misdiagnosed, there might be cases of unnecessary patient overtreatment [8]. Due to the limited experiences, no consensus or formal guideline for EBV-MCU treatment exist, and therapeutic strategies are only based on case reports and literature reviews.

If possible, immunosuppressive drug therapy should be reduced or discontinued as a first step. Conservative treatment is possible if the ulcer remains localised. In persistent or

progressive cases, rituximab monotherapy (375 mg/m$^3$, four weekly doses) or as a part of a chemotherapy (R-CHOP) should be considered. Local radiation had a beneficial effect in destructive courses. The selection of the appropriate therapy should be made in regard to clinical and individual aspects. However, in accordance with Hujoel et al., a lack of response within three months should always lead to another biopsy to re-evaluate the diagnosis of EBV-MCU [57].

## 6. Conclusions

Ulcerative lesions within the oral cavity or affecting the pharyngeal mucosa can have various reasons, and the correct diagnosis is difficult to make. The differentiation between more or less innocuous idiopathic, inflammatory or autoimmune disorders and malignant lesions, such as head and neck squamous cell carcinomas or lymphomas, are most decisive.

The wide range of various histological and immuno-histochemical features must be valued with caution, and the diagnosis of EBV-MCU has to be made considering the clinical aspects and in close collaboration between clinicians and pathologists. Misdiagnosis has a direct impact to patient care and may expose the patient to unnecessary risk.

Most cases of EBV-MCU are associated with age-related immunosenescence or iatrogenic immunosuppressive drug therapy. The majority present a self-limiting course without any treatment, and a dosage reduction of immunosuppressive drug therapy leads to complete remission in most cases. Therefore, aggressive therapy regimes can be especially harmful to an already compromised immune system and, in most cases, are unnecessary. Hence, they should only be used in non-regressing, progressing and relapsing cases.

While clear therapy recommendations can be made for the uncomplicated courses of the disease, this is not possible on the basis of the available studies for the non-regressing and progressing cases. On the one hand, this is due to the small number of patients who have received the individual aggressive therapies, but, above all, it is because no studies have been carried out to date that directly compare the different therapy options with each other.

The causes that lead to a severe course also remain unclear from the data available so far and require further investigation.

Therefore, further investigations are needed concerning those prolonged and more difficult cases.

At the same time, a precise definition of when a course of treatment is considered protracted and complicated remains open due to the inconsistent approach of the various studies and the strongly divergent (temporal) approach as do recommendations as to when an intensification of therapy should be started.

**Author Contributions:** Conceptualization, M.L. and P.B.; investigation, P.B. and M.L.; resources, P.B., M.L. and I.O.; writing—original draft preparation, P.B.; writing—review and editing, M.L.; visualization, P.B. and M.L.; supervision, A.R. and M.L. All authors have read and agreed to the published version of the manuscript.

**Funding:** This research received no external funding.

**Institutional Review Board Statement:** Ethical review and approval were waived for this study because it was not needed according to the guidelines.

**Informed Consent Statement:** Informed consent was obtained from all subjects involved in the study.

**Data Availability Statement:** Not applicable.

**Conflicts of Interest:** The authors declare no conflict of interest.

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
