# Peer review of "Co-Occurrence of EBV-Positive Mucocutaneous Ulcer (EBV-MCU) and CLL/SLL in the Head and Neck Region"

_curroncol, doi:10.3390/curroncol29040224_

Round 1

Reviewer 1 Report

The paper entitled "Co-occurrence of EBV-positive mucocutaneous ulcer (EBV-MCU) and chronic lymphatic leukaemia (CLL) in the head and neck region" is a review of the current literature and a recommendation for EBV-MCU management, plus a case report.

On the top left, please change "article" to "review" or "case report." 

Abstract: please, avoid acronyms (MCU) without explanations. 
Authors wrote:" All EBV-MCU cases of the head and 19 neck region published so far were analysed" and "A total of 145 cases were identified and 23 summarized.". However, the work is close to a systematic review but lacks its methods. I suggest authors should specify in the abstract that they investigated only PubMed. Otherwise, they should perform a complete and rigid systematic review of its features and methods. 

Furthermore, from line 176, the authors reported, "The PubMed database was searched from January 1, 1993 to December 18, 2020". We are now in April 2022, and the search should be updated. 
Furthermore, the search strategy need to be fully defined: did the authors use mesh terms? Inclusion/esclusion criteria? Please, be more precise. 
Due to the in-depth description of cases from the literature, I suggest authors perform a systematic review by including other databases, criteria of selection, and risk of bias assessment. 
Otherwise, they should discuss the literature, avoiding reporting the search strategy (pubmed..) . 

Due to the hybrid nature of the manuscript, both a case report and a review, I suggest dividing into different sections the two parts. 
A suggestion could be as follows:
Introduction
Case report
M&M (of the review )
Results (of the review)
Discussion (of the review and the case report)
Conclusions 

This is the simplest way to improve the quality of the writing. 
Otherwise, authors should consider performing a systematic review. 

However, the work is exciting and promising. 
I hope the authors will follow the suggestion to make it more suitable for publication. 

Author Response

Dear Reviewer, 

we have done our best to utterly implement your proposed changes to the manuscript. 

On the top left article was changed to Case Report and Review. 

Acronyms in the abstract were explained. The fact that only PubMed was investigated was added in the abstract. 

The search was updated and the additional cases published until April were added. That all published cases of EBV MCU in the head and neck area were included without any exclusion criterias was specified. 

The structure of the manuscript was changed following the suggestions. 

Reviewer 2 Report

The authors aimed to raise awareness of the disease, give a review of the current literature and a recommendation for EBV-MCU management. They also report a case of a pharyngeal EBV-MCU in an 89-year-old patient immunosuppressed by a chronic lymphatic leukaemia. In contrast to all previously described cases, histopathology showed a co-infiltration of EBV-MCU and CLL. A

The study covers some issues that have been overlooked in other similar topics. The structure of the manuscript appears adequate and well divided in the sections. Moreover, the study is easy to follow, but few issues should be improved. Some of the comments that would improve the overall quality of the study are:

  1. Authors must pay attention to the technical terms acronyms they used in the text. Please better stated the aim of the study in the introduction section.
  2. English language needs to be revised.
  3. Limitations of the study needs to be added.
  4. Conclusion Section: This paragraph required a general revision to eliminate redundant sentences and to add some "take-home message".

Author Response

1. Explanations of technical terms acronyms were added and the aim of the study was explaindes more thoroughly in the introduction. 

2. The English language was revised.

3. Limitations of the study were added in the conclusion section. 

4. The conclusion section was revised and formulated  as a short take home message. 

Reviewer 3 Report

In this article, a large number of documented cases of EBV-MCU in the head and neck region are comprehensively reviewed, and detailed treatments are reported and discussed. The paper also reports an interesting case of co-occurence of EBV-MCU and CLL, which is very rare. I think the paper can be published after revising the following minor points.

Minor points

  1. Page 2, line 63

The full spell of Epstein-Barr virus is shown here, but it should be noted in the first part of the Introduction.

  1. Page 2, line 87-91

Although classic Hodgkin lymphoma is indeed genetically derived from B-cells, it does not seem reasonable to include Hodgkin lymphoma in "B cell lymphoproliferation" for classification reasons. Similarly, some immunodeficiency-associated LPDs can histologically appear as Hodgkin lymphoma or, very rarely, peripheral T-cell lymphoma, so it may be inappropriate to include them in the "B-cell lympoproliferation".

  1. Page 3, line 139

Please define ENT before using the abbreviation.

  1. Figure 3

For figures c and d in Figure 3, it is not clear which cells are positive because of the low magnification, so please add a higher magnification histological image that shows the positivity for pleomorphic large cells.

  1. Figure 4

For figures b and c in Figure 4, it is not clear which cells are positive for CD23 and CD20 because of the low magnification. Please add a higher magnification image that shows both markers are positive on the cells of the same morphology.

Did you stain CD5 in the hypopharynx specimen? If so, it would be more informative with a histological image showing a similar distribution of CD5 and CD23.

  1. Please revise the typo "leucemia" in the legend of Figure 4.

  1. Please confirm that the magnification of CD30 (40x) shown in the legend in Figure 5 is correct.

  1. Since this case shows lymph node and organ involvement, CLL/SLL (small lymphocytic lymphoma) seems to be the proper description except for the initial presentation. Please check throughout the manuscript.

Author Response

1. The full spell of Epstein-Barr virus was changed to the first part of the Introduction.

2. The Hodgkin lymphoma and the immunodeficiency-associated LPDs were eliminated from the list of B cell lymphoproliferations

3. The abbreviation ENT was defined.

4. and 5. Unfortunately the pathologist in charge is on holiday for the whole week. But we will submit the requested images with higher magnification as soon as possible. 

6. The typo of leukemia was revised. 

7. The magnification was changed to the correct magnification of 400. 

8. The description was changed to CLL/SLL where appropriate

Round 2

Reviewer 1 Report

Dear Authors,

the quality of presentation has been notably improved.